# Human norovirus targets enteroendocrine epithelial cells in the small intestine

Kim Y. Green [1✉], Stuart S. Kaufman[2], Bianca M. Nagata[3], Natthawan Chaimongkol[1], Daniel Y. Kim[1], Eric A. Levenson [1], Christine M. Tin[1], Allison Behrle Yardley[1], Jordan A. Johnson [1], Ana Beatriz F. Barletta[4], Khalid M. Khan[2], Nada A. Yazigi[2], Sukanya Subramanian[2], Sangeetha R. Moturi[2], Thomas M. Fishbein[2], Ian N. Moore[3,5] & Stanislav V. Sosnovtsev[1,5]

Human noroviruses are a major cause of diarrheal illness, but pathogenesis is poorly understood. Here, we investigate the cellular tropism of norovirus in specimens from four immunocompromised patients. Abundant norovirus antigen and RNA are detected throughout the small intestinal tract in jejunal and ileal tissue from one pediatric intestinal transplant recipient with severe gastroenteritis. Negative-sense viral RNA, a marker of active viral replication, is found predominantly in intestinal epithelial cells, with chromogranin A-positive enteroendocrine cells (EECs) identified as a permissive cell type in this patient. These findings are consistent with the detection of norovirus-positive EECs in the other three immunocompromised patients. Investigation of the signaling pathways induced in EECs that mediate communication between the gut and brain may clarify mechanisms of pathogenesis and lead to the development of in vitro model systems in which to evaluate norovirus vaccines and treatment.

[1] Caliciviruses Section, Laboratory of Infectious Diseases, National Institute of Allergy and Infectious Diseases, National Institutes of Health, Bethesda, MD, USA. [2] MedStar Georgetown Transplant Institute and Georgetown University Medical Center, Washington, DC, USA. [3] Infectious Disease Pathogenesis Section, Comparative Medicine Branch, National Institute of Allergy and Infectious Diseases, National Institutes of Health, Bethesda, MD, USA. [4] Mosquito Immunity and Vector Competence Section, National Institute of Allergy and Infectious Diseases, National Institutes of Health, Bethesda, MD, USA. [5] These authors contributed equally: Ian N. Moore and Stanislav V. Sosnovtsev. ✉email: kgreen@niaid.nih.gov

Noroviruses, positive-sense single-stranded RNA viruses classified in the family *Caliciviridae*, are highly infectious human pathogens with a global distribution. Clinical outcomes of infection range from severe, life-threatening gastroenteritis to asymptomatic shedding of virus in stool. Moreover, noroviruses can establish chronic infection of varying severity in patients with underlying immune deficiencies[1]. Norovirus vaccines and therapeutics are a recognized public health need, but the development of such strategies has been hampered by poor understanding of pathogenesis in the human host.

Experimental model systems in animals have largely informed investigations of norovirus pathogenesis, but no animal model has fully replaced human challenge studies[2]. Murine norovirus (MNV) has been studied most extensively[3], with macrophage and dendritic cells bearing the virus receptor CD300lf defined as major target cells in mice[4]. Supporting this, MNV replicates in primary macrophage and dendritic cells cultured in vitro as well as in immortalized macrophage-derived cell lines such as RAW264.7[5]. Upon entry into the lamina propria via M cells[6], MNV replicates in both myeloid and lymphoid immune cells, including T and B cells in the gut-associated lymphoid tissue (GALT)[7]. Permissive cells in mice have recently been extended to include chemosensory tuft epithelial cells, the only known epithelial cells that express the CD300lf receptor[8].

Target cells for human norovirus replication are less well defined. Active replication has been linked to human enterocytes in the epithelial layer by co-localization of viral structural and nonstructural protein expression within the same cell[9]. Replication of human norovirus in an in vitro stem cell-derived intestinal enteroid cell culture system supported the identification of enterocytes as a permissive target cell[10]. Tissue from gnotobiotic piglets and calves challenged with human norovirus and bovine norovirus, respectively, showed the presence of viral antigen-positive enterocytes localized to the sides and tips of small intestinal villi[11,12]. Morphological changes to villi (blunting), including epithelial cell apoptosis, have been documented[13–15]. Human norovirus antigen has been detected in various immune cells of the lamina propria in both humans and animal models[9,12,16,17]. Evidence of replication in a human B cell line (BJAB)[18] prompted a search for B cell targets, but the role of immune cells in replication has not been verified in vivo[19].

The purpose of this study is to identify target cells in the human enteric tract that support active norovirus replication. Biopsies obtained from different portions of the intestine of a pediatric intestinal transplant patient with acute norovirus diarrhea are analyzed for the presence of viral antigen and RNA. Moreover, the identities of norovirus-infected cells are investigated with antibodies directed against cell-specific surface markers. In contrast to the striking tropism of MNV for immune cells[5,7], we show that human noroviruses target epithelial cells in this immunocompromised patient. Enteroendocrine cells (EECs), specialized epithelial cells in the intestine that secrete hormones and other signaling molecules integral to gut function, are identified as permissive target cells during acute and chronic infection

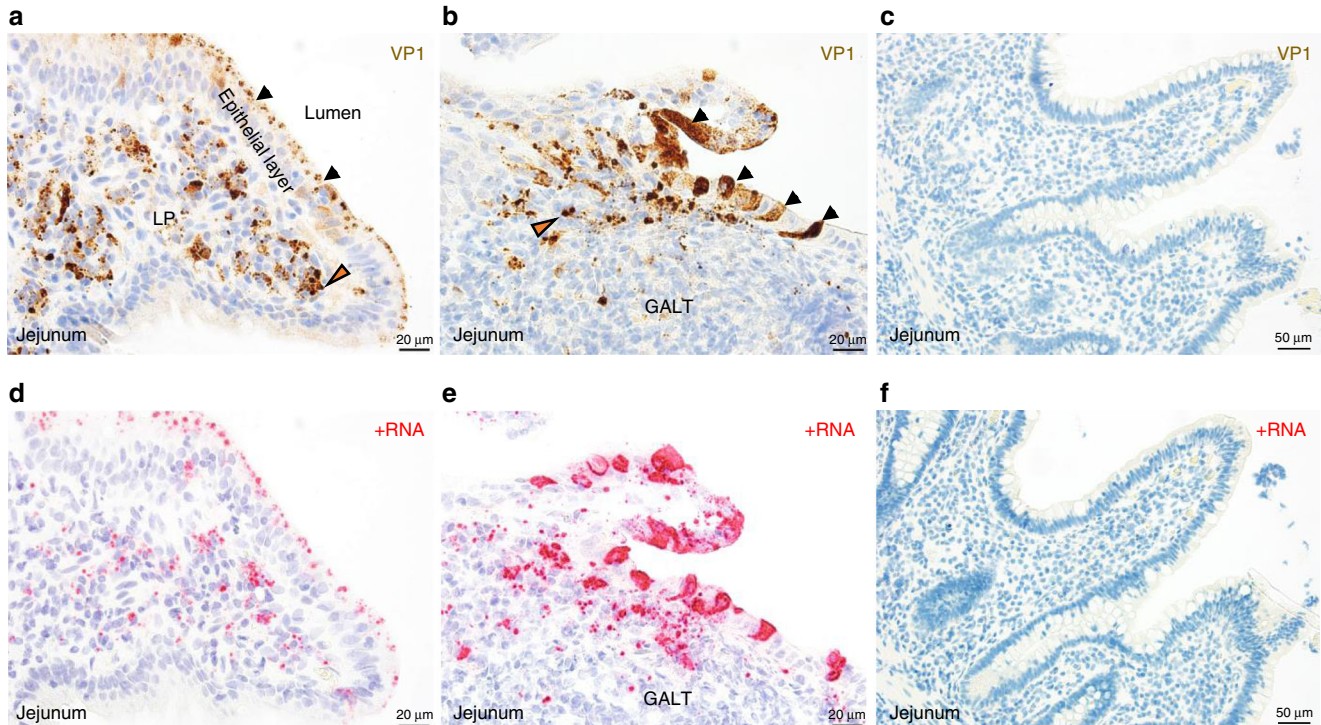

**Fig. 1 Detection of norovirus in jejunal biopsy from pediatric intestinal transplant patient (GT-1). a** Chromogenic staining and brightfield imaging with monoclonal antibody TV19 directed against the viral capsid protein (VP1) showed the presence of epithelial cells with an apical distribution of capsid antigen (staining brown) oriented toward the lumen (black arrows) as well as capsid-positive cells in the lamina propria (LP) (orange arrow). **b** Chromogenic staining of a different section of the same jejunal biopsy illustrated the presence of intensely stained epithelial cells above GALT (black arrows) as well as VP1-positive cells within the LP around the periphery of the GALT (orange arrow). **c** Chromogenic staining of jejunal biopsy from a norovirus-negative intestinal transplant patient (GT-4) with the TV19 capsid-specific monoclonal antibody was included as control. **d** In situ hybridization (ISH) analysis of patient GT-1 jejunal biopsy to detect positive-sense viral RNA (staining red) showed distribution patterns similar to VP1 in **a**, with an apical orientation of viral RNA toward the intestinal lumen in epithelial cells, as well as virus-positive cells within the LP. **e** ISH analysis of jejunal biopsy to detect positive-sense RNA showed a distribution pattern similar to VP1 in **b**, with epithelial cells above the GALT and in the LP on the periphery of GALT displaying positive reactivity. **f** Positive-sense RNA probe hybridized with jejunal biopsy from norovirus-negative transplant patient (GT-4) as control. (Magnifications: **a, b, d** and **e** 40×; **c** and **f** 20×).

in four immunocompromised patients. This opens avenues of investigation into the mechanisms of norovirus pathogenesis in diarrheal disease and in the development of optimized in vitro cell culture systems for the evaluation of vaccines and treatment.

## Results

**Norovirus is abundant in small intestine during acute disease**. A pediatric patient, designated here as Georgetown (GT)−1, presented with decompensated cirrhosis of unknown cause within the first months after birth and complicated by complete portomesenteric venous thrombosis precluding transplantation of the liver alone. At 9 months old the patient underwent multi-visceral transplantation[20] consisting of an *en bloc* stomach, duodenum, pancreas, liver, small intestine, and large intestine graft with splenectomy. Fourteen months later the patient presented with sudden profuse, nonbilious vomiting followed by equally sudden and profuse watery yellow-green diarrhea a few hours later resulting in severe dehydration. Gastrointestinal pan-endoscopy was performed 26 h after onset of vomiting and showed normal appearance of the duodenum-upper jejunum portion of the intestinal graft and the graft colon (Supplementary Fig. 1a, c), but abnormal terminal ileal graft with diffuse villous atrophy with red petechiae (Supplementary Fig. 1b). The patient was on maintenance immunosuppression with tacrolimus and mycophenolate mofetil accompanied by low-dose prednisolone that had followed induction immunosuppression consisting of basiliximab and high-dose corticosteroids[21]. At the time of illness, the patient's total lymphocyte count was approximately 4000–5000 cells/mcL, which was within normal range. Stool testing in a multiplex gastrointestinal pathogens diagnostic assay demonstrated only norovirus RNA genome, subsequently characterized as GII.4 Sydney[P16] norovirus at a titer of $2.72 \times 10^9$ genome copies per gram of stool.

Immunohistochemical chromogenic staining of the sectioned biopsies with a norovirus VP1-specific monoclonal antibody (TV19) showed the presence of abundant viral capsid antigen (staining brown) in localized regions of intestinal tissue (Fig. 1a and Supplementary Fig. 1d–f). Three general VP1 staining patterns were observed. First, certain virus-positive epithelial cells (morphologically consistent with enterocytes) showed a clear accumulation of punctate-appearing VP1 antigen along the inner apical surface oriented toward the lumen (Fig. 1a, black arrows). The second pattern was present in virus-positive cells within the lamina propria that displayed punctate or aggregated forms of VP1 antigen within cells (Fig. 1a, orange arrow). The third pattern was an intense, diffuse cytoplasmic distribution of viral capsid antigen within select epithelial cells that were distributed in a periodic fashion throughout the epithelial layer or grouped together near tips of villi or above an area of GALT (Fig. 1b, black arrows). Hybridization of an RNAscope probe designed to detect GII.4 norovirus positive strand RNA showed the presence of positive-sense RNA (staining red) in distribution patterns similar to that of the VP1 protein within the same area of tissue (Fig. 1d, e). Intestinal biopsies from norovirus-negative patients did not react with the VP1-specific monoclonal antibody TV19 (Fig. 1c) or the positive-sense norovirus RNA probe (Fig. 1f and Supplementary Fig. 2a). Norovirus VP1 and positive-sense RNA were not abundant in the colonic biopsy.

**Markers of active human norovirus replication in host cells**. An RNAscope probe was synthesized to detect the negative-strand RNA replicative intermediate that would be produced during active replication of the infecting GII.4 norovirus. Hybridization of the negative-sense RNA probe with a norovirus-negative biopsy confirmed the absence of background signal

(Supplementary Fig. 2b). The distribution of positive and negative-sense RNA in the jejunal tissue was compared by confocal fluorescence imaging with epithelial cell marker cytokeratin (CK) and macrophage marker IBA-1 to distinguish between the epithelial layer and the lamina propria, respectively. Similar to the VP1 and positive-sense RNA distribution patterns shown by IHC and brightfield microscopy in Fig. 1, viral positive-sense RNA was detected in both the epithelial layer and the lamina propria by confocal imaging (Fig. 2a–c). The three most commonly observed positive-sense RNA probe patterns were similar to those of VP1 (as noted above), with a punctate appearance and apical orientation in some epithelial cells (Fig. 2a, white arrow), aggregated-appearing forms in cells throughout the lamina propria (Fig. 2b, white arrow), and a diffuse cytoplasmic distribution in certain epithelial cells (Fig. 2c, white arrow). The viral negative-sense RNA signal was less abundant overall, likely reflecting its lower copy number during the norovirus replication cycle[22] (Fig. 2d–f). The negative-sense RNA probe pattern in tissue characteristically appeared as small discrete dot-like foci (Fig. 2e, f) or larger aggregated foci (Fig. 2d, e). One such cell bearing an aggregated negative-strand RNA signal (Fig. 2e, white arrow) was shown by 3D imaging to contain cytokeratin (Supplementary Movie 1), suggesting an epithelial origin. A panel of hyperimmune sera was raised in rabbits against GII.4 nonstructural proteins NS5$^{VPg}$, NS6$^{Pro}$, and NS7$^{Pol}$ to further define target cells for replication. The distribution pattern of these proteins in tissue was similar to that of the negative-strand RNA, with the strongest signals detected in or near the epithelium (Supplementary Fig. 3a–c, e). Cells co-expressing both negative-sense RNA and NS7$^{Pol}$ were observed within the epithelial layer (Supplementary Fig. 4).

The similar distribution patterns of positive-sense RNA and VP1 in the biopsy visualized by chromogenic staining (Fig. 1) prompted us to examine the relationship between VP1 and viral RNA by confocal fluorescence microscopy (Fig. 3 and Supplementary Fig. 5). Again, VP1 (green) and positive-sense RNA (red) were detected both in the epithelial layer and throughout the lamina propria (Fig. 3a and Supplementary Fig. 5a). Staining intensities and distribution of the VP1 and positive-sense RNA varied within the tissue. Co-localization of VP1 and positive-sense RNA was clear in a number of epithelial cells (Fig. 3a, cross-sections, yellow arrows and Supplementary Fig. 5b, inset white arrow). Again, the negative-sense RNA probe was less abundant and detected predominantly in or near cells of the epithelial layer (Fig. 3b and Supplementary Fig. 5c, d). The negative-sense RNA could be seen to co-localize with VP1 within certain epithelial cells (Fig. 3b, cross-section, yellow arrows), with the distribution pattern of negative-sense RNA and VP1 similar to that of the positive-sense RNA within certain epithelial cells (Supplementary Fig. 5d, inset white arrow). Overall, negative-sense norovirus RNA was difficult to visualize in the lamina propria, with some exceptions (Supplementary Fig. 5c, yellow arrow).

**Specialized epithelial cells and norovirus infection**. Intestinal epithelial cell types are diverse in morphology and function[23], and we investigated the identity of norovirus-positive epithelial cells that did not morphologically resemble absorptive enterocytes. We screened antibodies directed against known markers for various specialized epithelial cells and selected Glycoprotein 2 (GP2), choline acetyltransferase (ChAT), and chromogranin A (CgA) as optimal for the identification of M cells, tuft cells, and enteroendocrine cells (EECs), respectively (Supplementary Fig. 6a–c). Of these markers, only the EEC marker, CgA, could be consistently associated with the presence of norovirus negative-sense RNA and viral protein in scattered individual cells by confocal microscopy (Fig. 4a–c, Supplementary Fig. 7a–c, and

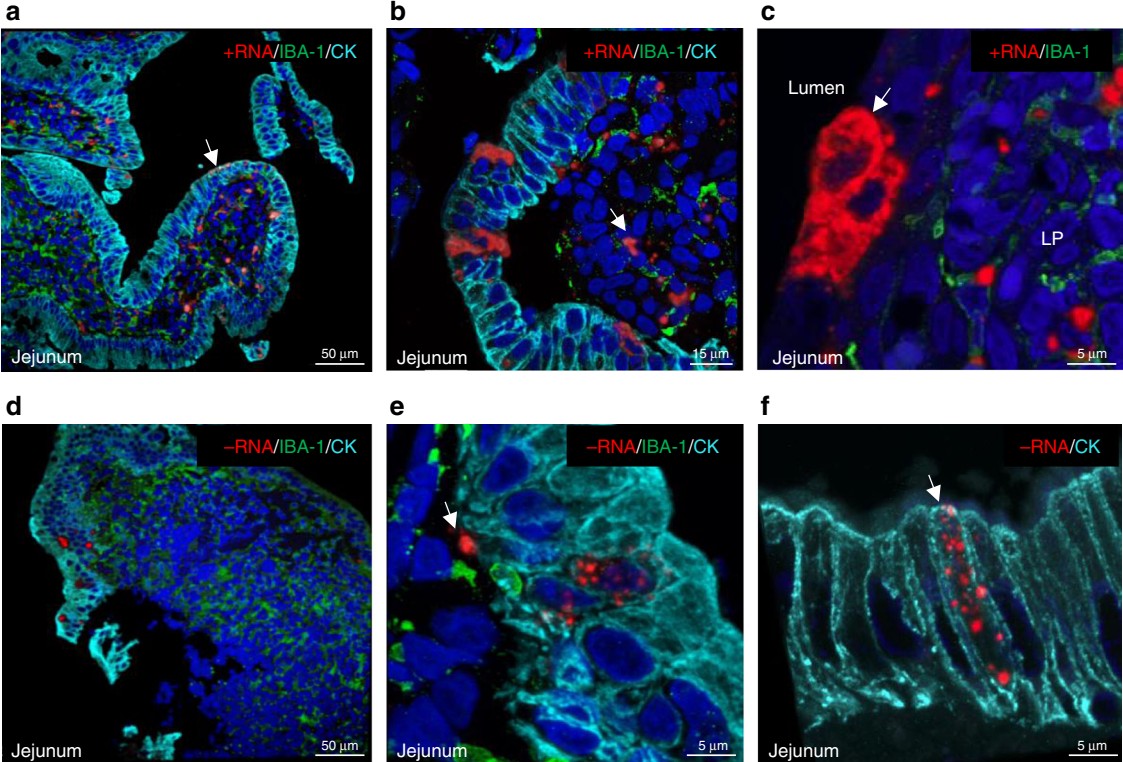

**Fig. 2 Positive and negative-sense norovirus RNA differ in distribution and intensity in GT-1 jejunal biopsies.** The GT-1 biopsy was analyzed by ISH with RNAscope norovirus-specific probes to detect: **a–c** positive strand RNA (+RNA) or **d–f** negative-strand RNA (-RNA). Characteristic distribution patterns of the different RNA species (red) are highlighted by white arrows and described in the text. The negative-sense RNA signal marked by the arrow in panel **e** was rendered in a 3D format with the Imaris Software package (Supplementary Movie 1). Cells in the epithelial layer were visualized by antibodies to detect cytokeratin (CK) (cyan) and those in the LP were represented by the detection of macrophage with marker IBA-1 (green). Nuclei are stained with DAPI (blue).

Supplementary Movies 2 and 3) and in additional patients (Supplementary Fig. 8a, b). The EECs are estimated to comprise approximately 1% of the intestinal epithelial cell population[24], and we determined the ratio of infected versus uninfected EECs in representative fields of view from three different patients (Supplementary Fig. 9). The percentage of infected EECs in each field varied from 3.5 to 25%, with an overall average of approximately 12%. We did not find evidence for active replication in ChAT-positive tuft cells, although these cells were present throughout the jejunal, ileal, and colonic tissue. GP2-positive cells were scattered throughout the epithelium of patient GT-1, but the paucity of remaining GALT tissue for analysis with the negative-sense RNA probe made it difficult to establish the role of M cells at this time.

**Myeloid immune cells and norovirus infection.** We investigated human norovirus replication in myeloid immune cells. The biopsies were probed for expression of DC-SIGN, a type C lectin receptor present on the surface of both macrophage and dendritic cells (Fig. 5). Association of the DC-SIGN marker (red) with norovirus VP1-positive cells (brown) was found throughout the small intestinal lamina propria of patient GT-1 by chromogenic staining, with varying amounts of punctate or aggregated viral antigen within cells (Fig. 5a). In a domed area above the GALT, DC-SIGN positive cells containing norovirus VP1 appeared to accumulate between the virus-positive epithelium and the mostly virus-negative lymphoid tissue (Fig. 5b). Confocal microscopy confirmed the association of VP1 and DC-SIGN (Supplementary Fig. 10, white arrow and Supplementary Movie 4). Similar staining patterns were observed with IBA-1, a marker expressed

on both activated and resting macrophage (Fig. 5c, d). Capsid VP1 (brown) co-localized with IBA-1 (red)-positive cells within the lamina propria, and in some fields, appendage-like dendritic structures bearing viral antigen were extended toward the epithelial cell barrier (Fig. 5c, inset arrows). An interaction of IBA-1 positive macrophage with capsid-positive cells in the epithelial layer was observed in some areas (Fig. 5d, inset arrows). Confocal microscopy confirmed co-localization of cytokeratin and viral capsid antigen within certain IBA-1 positive cells in the lamina propria, consistent with active scavenger activity during the acute infection (Supplementary Fig. 11). In other areas of tissue, capsid protein was present in the lamina propria in the absence of abundant cytokeratin and without clear proximity to infected epithelial cells (Supplementary Fig. 12). We investigated the presence of negative-sense RNA in macrophage by immunofluorescence techniques (Fig. 5e, f). As noted above, negative-sense RNA was characteristically observed in the epithelial layer, and it was difficult to find such cells for 3D imaging in the lamina propria.

**Lymphoid immune cells and norovirus infection.** Because MNV has been reported to replicate in B and T cells, we next examined whether human norovirus VP1 antigen was present in these lymphoid immune cells (Fig. 6). An analysis of GALT probed for positive-sense norovirus RNA (Fig. 6a, expanded from Fig. 1e) showed an abundance of both B (CD20-positive) (Fig. 6b) and T (CD3-positive) (Fig. 6c) cells in the tissue, as expected. Notably, the innermost region of the GALT consistently lacked viral RNA and VP1 signals throughout the study, although such signals were readily observed around the periphery of GALT in

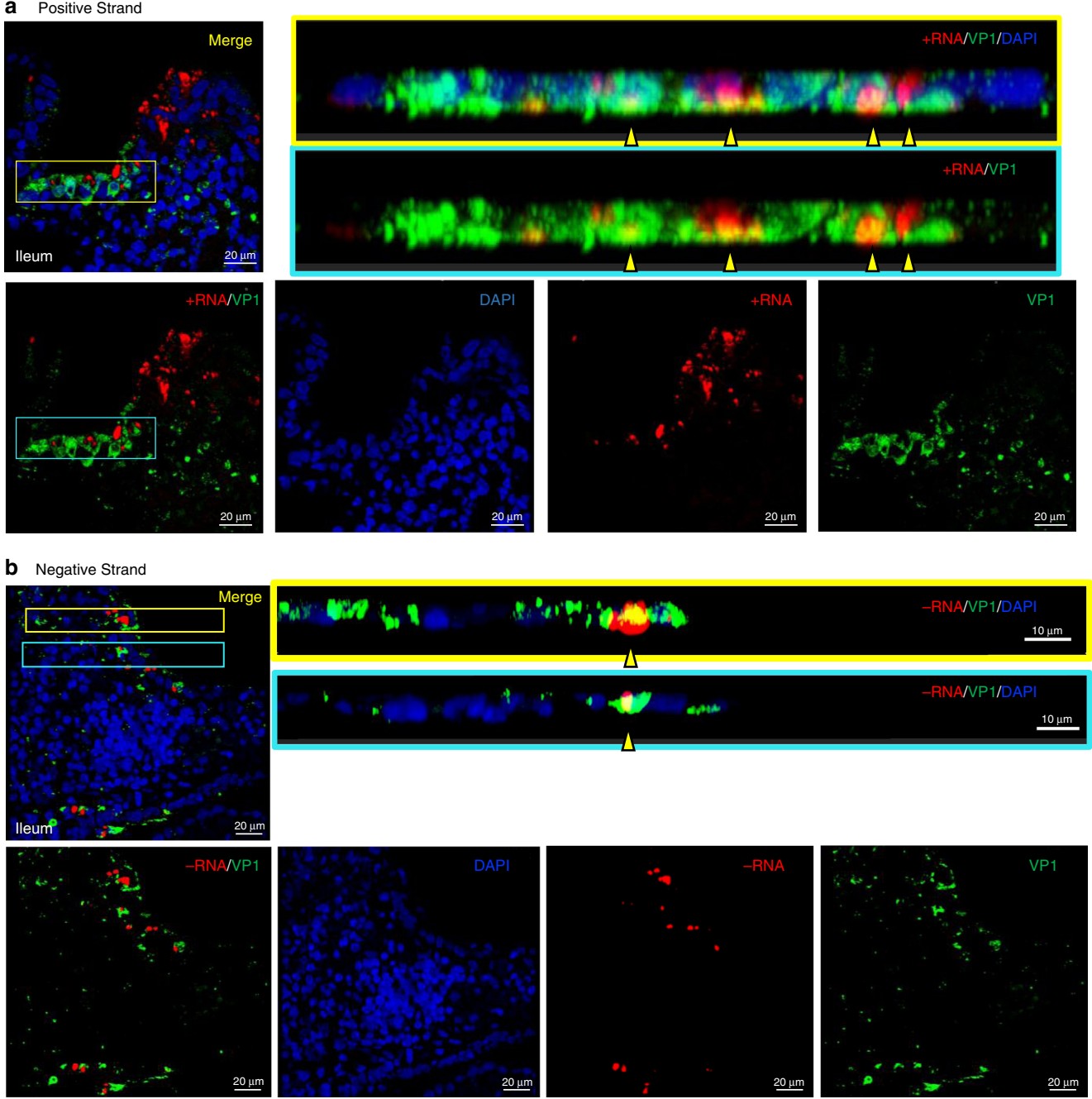

**Fig. 3 Association of capsid VP1 expression with positive or negative-sense viral RNA. a** Confocal microscopy imaging of VP1 (green, visualized with TV19) and positive-sense RNA (red, hybridized with positive RNAscope probe) in the GT-1 ileal biopsy. Merged and separate staining patterns for each marker are shown. Side views of the merged images show overlap of positive-sense RNA and VP1 expression (indicated by yellow arrows) in the presence or absence of DAPI staining. **b** Confocal microscopy imaging of VP1 (green, visualized with TV19) and negative-sense RNA (red, hybridized with negative RNAscope probe) in the GT-1 ileal biopsy. Merged and separate staining patterns for each marker are shown. The overlap between VP1 and negative-sense RNA is shown in two cross sections (yellow arrows) from proximal regions in the tissue (inset box outline colors correspond to highlighted regions in merged (**b**) image).

association with myeloid immune cell markers (Figs. 6a and 5b). There was no consistent association of the viral VP1 antigen (brown) with CD3-positive T cells (red) within the lamina propria or in subepithelial locations (Fig. 6d). Consistent with this, there was little evidence of association between VP1 and CD4-positive T helper cells (Fig. 6e) or VP1 and CD103-positive intraepithelial lymphocytes (IELs) (Fig. 6f). Confocal microscopy of a region of the GT-1 biopsy probed for CD4 and the norovirus positive-sense RNA showed possible overlapping signals, but further analysis illustrated that the RNA signal was on the periphery of the T cell membrane (Supplementary Movie 5). Likewise, the CD20-positive B cells found throughout the lamina propria and in lymphoid tissue did not co-localize or associate with virus antigen or RNA. Confocal imaging of the GT-1 biopsy with both T and B cells markers and the negative-sense RNAscope probe failed to detect co-localization within these immune cells in this study (Fig. 6g and Supplementary Movie 6).

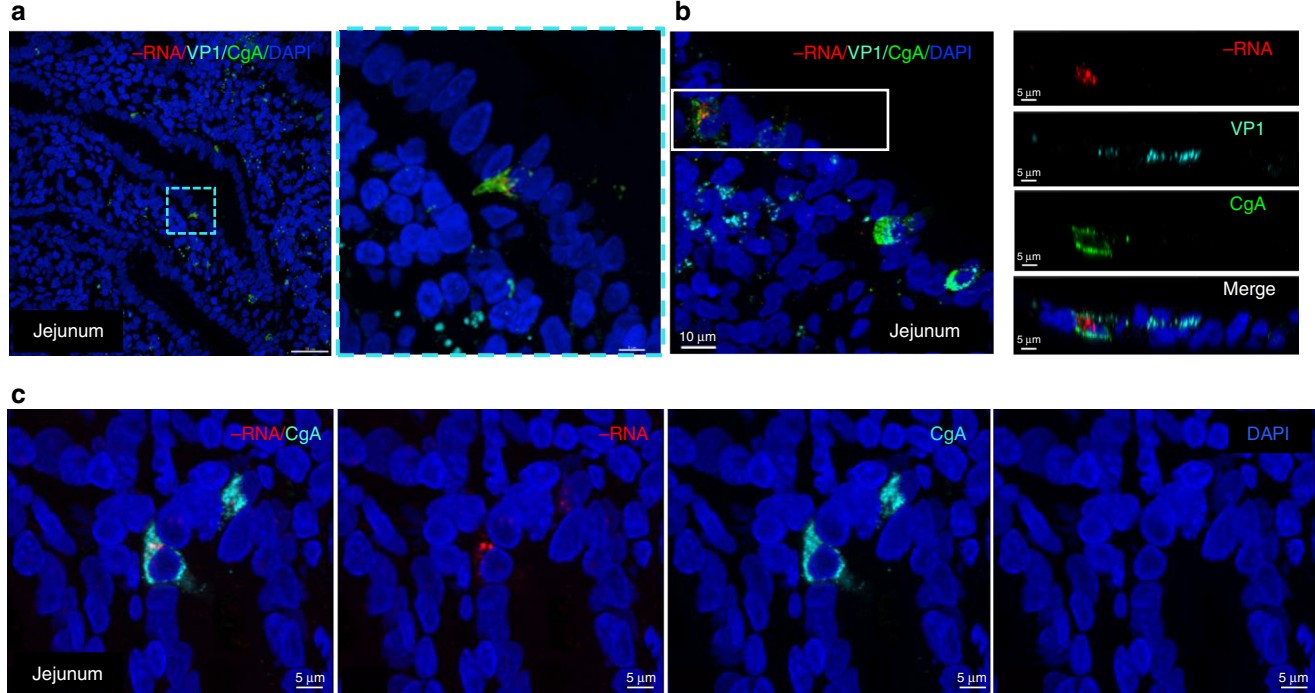

**Fig. 4 Presence of norovirus negative-sense RNA in chromogranin A-positive cells.** Chromogranin A (CgA) is a protein released by neuroendocrine cells, and it has been used as a general marker for intestinal enteroendocrine cells (EECs). **a** Confocal imaging of the GT-1 jejunal biopsy showing CgA (green)-positive cells with and without the presence of norovirus negative-strand RNA (red) and VP1 (cyan) in the intestinal epithelium. Inset shows magnified view of the norovirus-positive EEC. **b** Confocal imaging and side views of an individual cell staining with CgA (green), negative-sense RNA (red) and VP1 (cyan). **c** GT-1 jejunal biopsy with two proximal epithelial cells bearing the presence of both CgA and negative-sense RNA. Merged and separate images of norovirus negative strand (red), CgA (cyan), and DAPI (blue) nuclei staining are shown.

## Discussion

A pediatric intestinal transplant recipient presenting with severe norovirus gastroenteritis provided a rare opportunity to study norovirus infection during acute, life-threatening illness. Norovirus has been identified as a potential risk factor for acute secondary organ rejection in this patient population[25]. The intensity of protein and RNA staining in intestinal tissue was reflected in the high number of viral genome copies in stool, allowing us to investigate the distribution and cellular tropism of norovirus in the likely absence of pre-existing immunity. Norovirus was distributed throughout the small intestine, with concentrated areas of virus in epithelial cells at or near the tips of small intestinal villi and above regions of GALT. Enteroendocrine cells were identified as an epithelial cell type associated with active norovirus replication. Enteroendocrine cells are specialized epithelial cells with sensory and endocrine functions that comprise approximately 1% of the intestinal epithelium[26]. This discovery may offer insight into the pathogenesis of human norovirus, where symptoms often have an acute onset with explosive vomiting and diarrhea. Rotavirus, another enteric virus associated with gastroenteritis, was shown to infect enterocytes and secrete viral protein NS4, which caused the release of serotonin from proximal enterochromaffin cells (ECs), a subtype of EECs[27]. There was evidence also that ECs could occasionally be infected directly by rotavirus[27], and serotonin receptor antagonists have been explored for the treatment of rotavirus-associated vomiting and diarrhea[28]. Additional study will be needed to establish the role of EECs in both immunocompetent and immunocompromised human hosts with norovirus disease, but the wide-ranging roles of EECs in gut-brain signaling, digestion, and mucosal immunity could offer important insight into strategies to treat norovirus illness.

Our findings are consistent with a number of published observations in animals and humans, notably those from the analysis of intestinal biopsies from immunocompromised patients with chronic norovirus infection[9]. Previous observations, coupled with our strand-specific viral RNA analysis presented here (albeit in a limited number of immunocompromised patients), indicate that intestinal epithelial cells in the small intestine are major primary target cells for human norovirus replication (Supplementary Fig. 13). This differs from murine norovirus, which predominantly targets immune cells during primary infection[5,7,29]. Human norovirus is internalized into myeloid cells in the lamina propria by phagocytosis of debris from infected epithelial cells or by lysis-independent spread, perhaps via exosomes as proposed recently[30]. Whether internalized human norovirus remains viable and infectious within myeloid immune cells for a period of time or replicates under certain conditions could not be determined in our study, but it is likely that gut homeostasis is affected by the presence of large numbers of activated antigen-presenting cells, especially during an acute infection. Immune cell activation, coupled with a strong innate immune response and chemical signaling in specialized epithelial cells, including EECs, may contribute to the signs and symptoms of norovirus disease. The elucidation of host response pathways involved in norovirus pathogenesis and the identification and inclusion of permissive EEC subtypes in cell culture systems may enhance efforts to investigate antiviral therapies and vaccines for these viral pathogens.

## Methods

**Ethics statement.** Clinical studies were conducted with approval and oversight from the Georgetown University Institutional Review Board (IRB) and included written informed consent from patients and/or their legal representatives. Human biospecimens collection and analysis at NIH was approved by the NIH IRB and

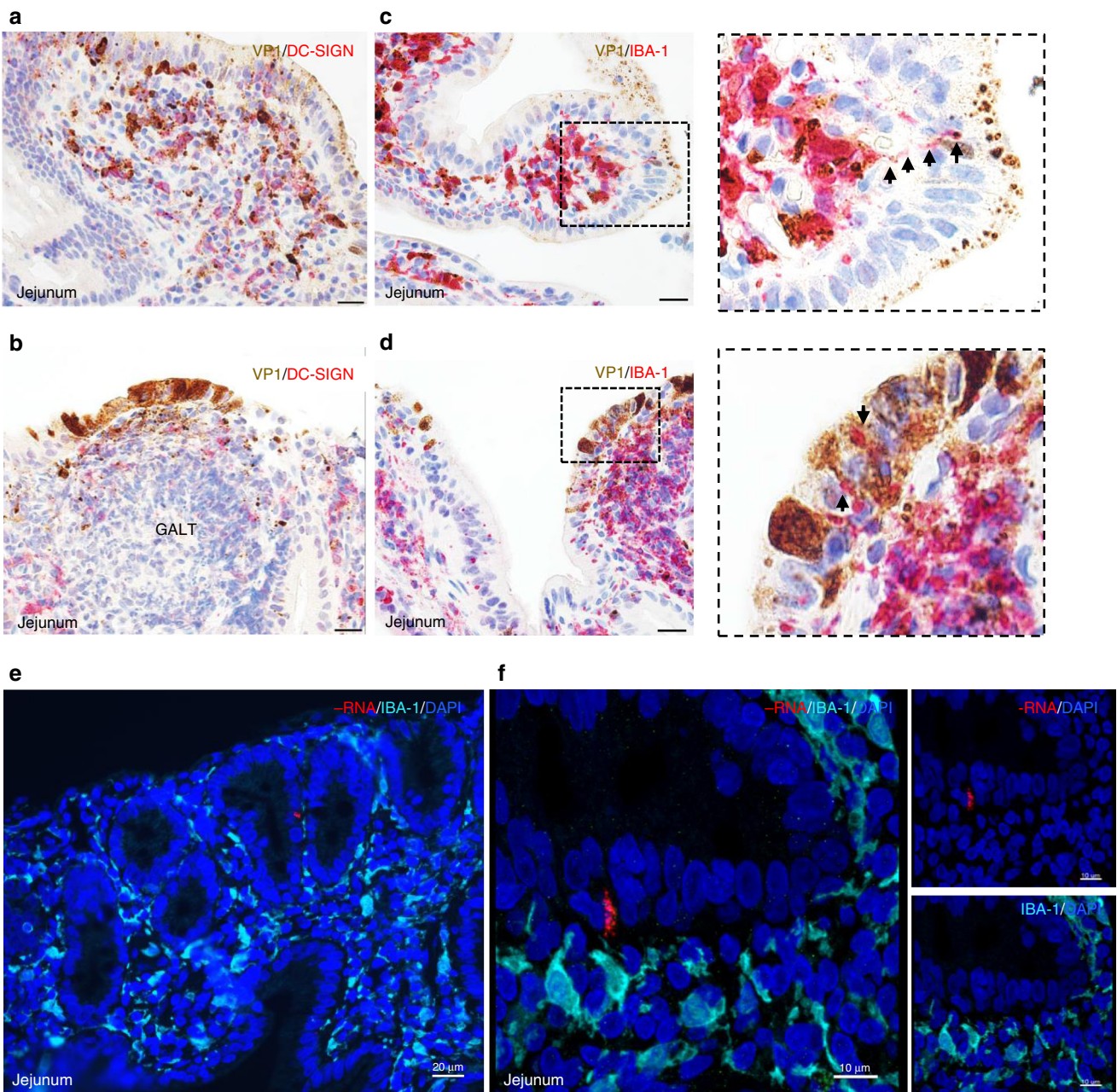

**Fig. 5 Association of norovirus VP1 capsid protein with myeloid immune cells in the lamina propria.** Chromogenic staining and brightfield imaging were used to show the relationship between VP1 and DC-SIGN bearing cells in two areas of the GT-1 small intestine. **a** Villus structure showing VP1 and DC-SIGN staining merged, with heavy presence of VP1 within the LP. **b** Area containing GALT showing VP1 and DC-SIGN staining merged. **c** View near villus tip shows association of VP1 with macrophage bearing IBA-1 in the LP. Magnified inset shows extension of a macrophage dendrite appearing to engulf VP1 antigen (marked by arrows). **d** An area of intense VP1 expression in the epithelial layer is shown with an accumulation of underlying macrophage containing aggregated VP1. Magnified inset highlights evidence for direct interaction of IBA-1 positive cells with the epithelium as indicated by arrows. (Magnifications: **a–d** 40×, scale bars represent 20 μm). **e** Immunofluorescence imaging of a section of the jejunum hybridized with the negative-sense RNA probe (red) in the presence of macrophage marker IBA-1 (cyan). **f** A different field of the section shown in **e** was visualized by confocal imaging, and an epithelial cell expressing negative-strand RNA (red) is shown proximal to a macrophage (cyan) in the lamina propria.

included written informed consent. Animal studies were approved by the Institutional Animal Care and Use Committee of an NIH-approved vendor and conducted adhering to the guidelines and basic principles in the United States Public Health Service Policy on Humane Care and Use of Laboratory Animals, and the Guide for the Care and Use of Laboratory Animals by certified staff in an Association for Assessment and Accreditation of Laboratory Animal Care (AAALAC) International accredited facility.

**Patient samples.** The samples from four norovirus-positive and two norovirus-negative patients analyzed in the present study were obtained under IRB-approved protocols at two participating medical centers, MedStar Georgetown Transplant Institute of the MedStar Georgetown University Hospital (Washington, D.C.) and the National Institutes of Health Clinical Research Center (Bethesda, MD).

**Norovirus quantification and genotyping.** Stool samples were screened initially with the BioFire FilmArray Gastrointestinal Panel (bioMérieux). Norovirus-positive stool was prepared as a 10% w/v suspension in PBS, and viral RNA was extracted from a 50 μl volume with the MagMAX-96 Viral RNA Isolation Kit (Thermo Fisher Scientific). The presence of norovirus RNA, presumptive genogroup (G) assignment, and viral genome copies per gram of stool were determined by reverse transcription quantitative polymerase chain reaction (RT-qPCR) with the TaqMan Fast Virus 1-Step Master Mix (Thermo Fisher

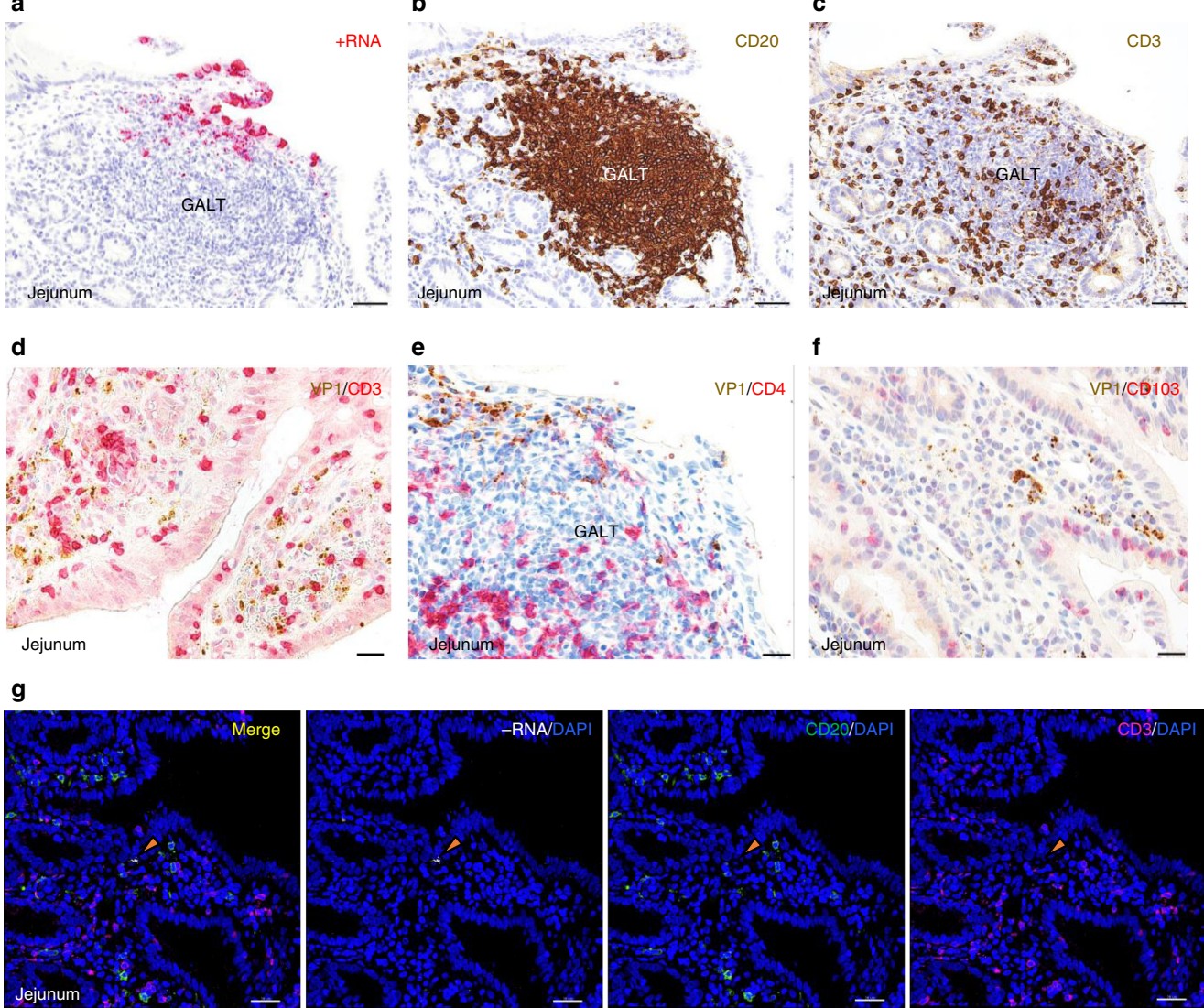

**Fig. 6 Relationship of human norovirus to B and T lymphoid cells.** The GT-1 jejunal biopsy was probed for expression of VP1 in cells expressing various immune cell markers by chromogenic staining and brightfield imaging. **a** Reproduction and wider view of the identical image in Fig. 1e showing the distribution of positive-sense norovirus RNA, included here for direct comparison with panels **b** and **c**. **b** The region of the biopsy containing GALT and norovirus positive cells shown in **a**. was probed only with B cell marker CD20 (brown), and the density of B cells within the GALT can be seen. **c** An area in proximity to **a** and **b** was probed only with T cell marker CD3 (brown) to demonstrate the expected presence of T cells in GALT. Additional areas of the GT1 jejunal biopsy were probed for both VP1 (brown) and: **d** T cell marker CD3 (red). **e** T helper cell marker CD4 (red) and **f** CD103, a marker for intraepithelial lymphocytes (red). **g** Confocal microscopy imaging showing merged and unmerged channels to detect norovirus negative-sense RNA (white), B cells (green), and T cells (magenta) in the presence of DAPI staining (blue). Location of the norovirus target cell is indicated with orange arrow in top panel, and 3D modeling (Supplementary Movie 5) of this cell showed the absence of co-localization with T cell marker CD3 (Magnifications: **a–c** 20×, scale bars 50 μm; **d–f** 40×, scale bars 20 μM; **g** 20×, scale bar 30 μm).

Scientific) according to the manufacturer's instructions. Primer pair MGBGII_F/MGBGII_R and probe MGBGII[31] were used in conjunction with a synthesized RNA standard (Bio-Synthesis) representing the consensus sequence of a highly conserved region in NS7[Pol] to calculate viral genome copy numbers (Supplementary Table 1). For genotype determination, cDNA was generated using the Quanta qScript cDNA Synthesis Kit (Quantabio) and a reverse primer G2SKR[32] that targets the GII capsid region (open reading frame 2, ORF2) (Supplementary Table 1). cDNA was used as template for a semi-nested PCR protocol (Supplementary Table 1) to generate amplicons of approximately 700 bp at the ORF1/2 junction using the Platinum SuperFi DNA polymerase (Invitrogen). PCR products were gel purified and the nucleotide sequences were determined directly from the amplicons using the Big Dye Terminator Cycle Sequencing Ready Reaction Kit, version 3.0 (Applied Biosystems) and an ABI PRISM 3730 automated DNA analyzer (Applied Biosystems). An online norovirus typing tool[33] in conjunction with NCBI BLAST was used to assign the norovirus genotype. Once the genotype was assigned, genotype-specific primers were used to amplify and sequence the full-length ORF2 (Supplementary Table 1).

The GT-1 sequence determined in this study was deposited in GenBank and assigned Accession Number MN220720.

**Histological procedures.** Endoscopic biopsies collected from jejunum, ileum, and colon were fixed in 10% neutral buffered formalin prior to embedding and processing in histological grade paraffin. All tissues were sectioned at 5 μm and stained with hematoxylin-eosin (H&E) for examination by light microscopy. Sections received at NIH were evaluated by a board-certified veterinary pathologist. Images were taken using an Olympus BX51 microscope (Olympus Corp.) and photomicrographs were taken using an Olympus DP73 camera.

**Pretreatment for staining on formalin fixed, paraffin embedded (FFPE) tissues.** All chromogenic immunohistochemistry (IHC), immunofluorescence (IF), and in situ hybridization staining was performed on the Leica Bond RX automated system. All tissue sections were dewaxed for 30 min in Bond Dewax Solution (Leica Microsystems) heated to 72 °C. Tissues were then rehydrated with absolute ethanol

washes and 1x ImmunoWash (StatLab). All antibodies were diluted in Background Reducing Antibody Diluent (S3022, Agilent).

**Single or double brightfield immunohistochemistry (IHC).** Epitopes were retrieved through a 20 min treatment in Bond Epitope Retrieval Solution 1 (Leica Microsystems) heated to 100 °C. Only slides to be detected with 3,3′-Diamino-benzidine (DAB) tetrahydrochloride hydrate were exposed to peroxide block (Leica Microsystems) for 5 min. Sections were incubated with the following primary antibodies for the detection of cellular markers (Supplementary Table 2): CD3 (Bio-Rad Laboratories; Clone: CD3-12, Catalog No: MCA1477, Dilution: 1:600 for IHC or 1:400 for immunofluorescence), CD4 (Abcam; Clone: EPR6855, Catalog No: ab133616, Dilution: 1:500), CD20 (Abcam; Clone: EP459Y, Catalog No: ab78237, Dilution: 1:100 for IHC), CD20 (Abcam; Clone: SP32, Catalog No: ab64088, Dilution: 1:100 for immunofluorescence), CD103 (Novus Biologicals; Catalog No: NBP1-88142, Dilution: 1:50), DC-SIGN (Abcam; Catalog No: ab5715, Dilution:1:500), IBA-1 (Wako; Catalog No: 019-19741, Dilution: 1:800 for IHC), IBA-1 (Abcam; Catalog No: ab107159, Dilution: 1:1000 for immuno-fluorescence), Pancytokeratin (CK) (Abcam; Clone: AE1/AE3, Catalog No: ab27988, Dilution: 1:200), GP2 (Novus Biologicals; Catalog No: NBP1-86061, Dilution: 1:1000), Chromogranin A (CgA) mouse monoclonal (Biocare Medical; Clone: LK2H10+PHE5, Catalog No: CM010, Dilution: 1:200 for IHC or 1:50 for immunofluorescence, Chromogranin A (CgA) rabbit polyclonal (Abcam; Catalog No: ab15160, Dilution: 1:400), and Choline acetyltransferase (ChAT) (Abcam; Clone: EPR16590, Catalog No: ab178850, Dilution: 1:2000 for IHC or 1:500 for immunofluorescence) (Supplementary Table 2). For detection of norovirus structural capsid protein VP1, monoclonal antibodies TV19[34] and 30A11[35] were used at dilutions of 1:100 and 1:500, respectively. For detection of norovirus nonstructural proteins, primers pairs were designed to amplify the NS5[VPg], NS6[Pro] and NS7[Pol] regions of GII.4 norovirus Rockville/2012 (GenBank KY424328), each pair with an incorporated histidine tag and restriction endonuclease sites for cloning into the pET28b vector (EMD Biosciences) (Supplementary Table 1). Recombinant proteins were overexpressed in *E. coli* BL21 (DE3) cells and purified by immobilized metal (nickel) affinity chromatography prior to immunization of rabbits[36]. Rabbit immunizations were conducted according to an Animal Care and Use Committee-approved protocol at Pocono Rabbit Farm and Laboratory Inc., and the resulting three nonstructural protein-specific antisera were each used at a dilution of 1:50. Following incubation with primary antibodies for 15 min, the sections were rinsed with ImmunoWash and treated with an appropriate secondary antibody if needed. Primary antibodies with a rat host were treated for 8 min with an unconjugated rabbit anti-Rat IgG antibody (Vector Laboratories; Catalog No: AI-4001, Dilution: 1:100). The Polymer Refine Detection System (Leica Microsystems) was used for DAB and the Polymer Refine Red Detection System (Leica Microsystems) for Fast Red detection. Colorization was achieved with DAB incubation for 10 min and Fast Red for 15 min. Counter-staining was achieved with hematoxylin. In the case of double IHC, the DAB protocol was applied first and was followed by Fast Red detection. Epitope retrieval was applied prior to the DAB protocol.

**Single and sequential immunofluorescence.** The epitope retrieval protocol for immunofluorescence (IF) was the same as that for IHC. Prior to primary antibody incubation, samples were blocked with Serum-Free Protein Block (X0909, Agilent) for 30 min. Samples were incubated in a primary antibody solution for 60 min then rinsed with ImmunoWash. Secondary antibodies (Supplementary Table 2) against the primary antibody host species were incubated in the tissues for 30 min. Primary antibodies were treated for 30 min with either a goat anti-rabbit secondary (Thermo Fisher Scientific; Catalog No: A11008, Dilution: 1:500), biotinylated horse anti-mouse (Vector Laboratories; Catalog No: BA-2000, Dilution: 1:200), or bio-tinylated donkey anti-mouse (Thermo Fisher Scientific; Catalog No: A16021, Dilution: 1:500). For double labeling, the second primary antibody was applied to the tissue for 60 min followed by a 30 min incubation of a secondary antibody against the host species of the second primary antibody. Primary antibodies with a mouse host were treated for 30 min with a biotinylated horse anti-mouse secondary (Vector Laboratories; Catalog No: BA-2000, Dilution: 1:200) which was followed by a 30 min incubation with a streptavidin conjugate (Thermo Fisher Scientific; Catalog Nos: S32354, S32356, S32358 each at 1:500 dilution). Slides were counterstained with DAPI (Thermo Fisher Scientific) at 1:1000 dilution in 1x PBS for 2 min.

**Simultaneous immunofluorescence.** Tissues were treated with the epitope retrieval protocol. Samples to be treated with a cocktail containing a biotinylated secondary were incubated with a streptavidin and biotin blocking solution (Vector Laboratories; Catalog No SP-2002). All tissues were blocked with Serum-Free Protein Block for 30 min. Samples were subsequently incubated in a cocktail mixture of primary antibody solution for 60 min then rinsed with ImmunoWash. A cocktail mixture of secondary antibodies against the primary antibody host species was applied to the tissues for 30 min. The secondary antibodies (Supplementary Table 2) that were used in the study include the following: donkey anti-rabbit (Thermo Fisher Scientific; Catalog No: A21206, Dilution: 1:1000), horse anti-rabbit (Vector Laboratories; Catalog No: DI-1088, Dilution: 1:300), biotiny-lated donkey anti-mouse (Thermo Fisher Scientific; Catalog No: A16021, Dilution:

1:500), biotinylated horse anti-mouse (Vector Laboratories; Catalog No: BA-2001, Dilution: 1:75), biotinylated donkey anti-rat (Novus Biologicals; Catalog No: NBP1-75379, Dilution 1:500), and biotinylated donkey anti-goat (Thermo Fisher Scientific; Catalog No: A16009, Dilution: 1:500). This was followed by an incubation with a streptavidin conjugate for 30 min. Slides were counterstained with DAPI.

**In situ hybridization (ISH) brightfield and immunofluorescence detection.** RNAscope probes (ACDBio) were synthesized to detect the positive-sense and negative-sense RNA species of the GII.4 ORF2, based on sequence of the related norovirus Hu/USA/2015/ GII.4 Sydney[P16]/Pasadena 3477 (GenBank Accession Number KY947550). PPIB Positive Control (ACDBio) and dapB Negative Control (ACDBio) were used as controls in select cases. Epitope retrieval was performed on tissues with application of Bond Epitope Retrieval Solution 2 (Leica Microsystems) at 88 °C for 15 min followed by a Protease III (ACDBio) treatment at 40 °C. Tissues were hybridized with the probes for 120 min and visualized with either the RNAscope 2.5 LS Reagent Kit-RED or the RNAscope LS Multiplex Fluorescent Assay (ACDBio). The detection kits allowed for visualization of the probe through chromogenic and/or fluorescence methods. For ISH coupled with IF, the slides were treated with the ISH procedure prior to IF. After the ISH protocol, sections proceeded with the IF protocol detailed above without an additional epitope retrieval step. Slides were counterstained with DAPI.

**Confocal microscopy.** Images were captured using a Leica TCS SP8 DMI6000 inverted fluorescence confocal microscope (Leica Microsystems) equipped with a photomultiplier tube/hybrid detector and a Leica DFC345 Monochrome camera. Images were viewed with the 20×, 40x or 63x oil immersion objective (zoom factor of 2, 3 or 4) and data were collected using the Leica Application Suite X software platform. White light laser and specific emission and excitation range were applied depending on the fluorophore. The following spectra for excitation/ emission were used: 488/520 for IBA-1, DC-SIGN, VP1 and ChAT; 594/620 for viral RNA positive and negative-sense probes; and 680/700 for CK and CgA. DAPI was excited using a 405-nm diode laser. Images were taken using sequential acquisition and variable z-steps. Image processing was performed using Imaris 9.2.1 (Bitplane, Oxford Instruments). Maximum intensity projection (MIP) is represented in xy pictures, and cross sections of between 5–7 μm were obtained with the ortho slicer feature from Imaris.

**Data analysis and reproducibility.** Captured images were prepared for presentation in Photoshop CC 2019 (Adobe, Inc.) and scale bar lengths were defined either directly on the figure or in the corresponding legend. Data are presented as representative images of multiple fields of view for each antibody or probe analyzed per biopsy section. Analyses were performed 3–5 times on each biopsy and staining patterns were reproducible within and between the four norovirus-positive patients in the study. Biopsy sections from two norovirus-negative patients were included as controls to monitor nonspecific signals with the antibodies and probes.

**Reporting summary.** Further information on research design is available in the Nature Research Reporting Summary linked to this article.

## Data availability

Sequence data that were generated in this study have been deposited in GenBank with the primary accession code MN220720 [https://www.ncbi.nlm.nih.gov/nuccore/MN220720]. All other data are available from the authors upon request.

## Code availability

No previously unreported custom computer code or algorithm were used to generate results.

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

## Acknowledgements

This work was funded by the Division of Intramural Research, NIAID, NIH, DHHS. We thank Dr Jeffrey I. Cohen, Chief, Laboratory of Infectious Diseases (LID), NIAID for his support and scientific advice. We thank Dr Juraj Kabat of the NIAID Biological Imaging Section for assistance with confocal imaging and Mr Ryan Kissinger of NIAID for the summary illustration in Supplementary Fig. 3. We thank Ms Courtney Ahorrio for management of the Caliciviruses Section, LID.

## Author contributions

K.Y.G. designed experiments, analyzed data and wrote manuscript, S.S.K. provided patient care, collected clinical samples, analyzed data and wrote manuscript, B.M.N. performed immunohistochemistry and in situ hybridization, N.C. and D.Y.K. characterized noroviruses in clinical samples, E.A.L. developed norovirus assays and designed experiments, C.M.T. generated region-specific norovirus antisera, A.B.Y. and J.A.J. performed norovirus analysis, A.B.F.B. collected and processed confocal microscopy images, K.M.K., N.A.Y., S.S., S.R.M., collected and analyzed biopsy specimens, T.M.F. led surgical transplant team and reviewed manuscript, I.N.M. and S.V.S. designed experiments, collected and analyzed data, and wrote manuscript.

## Competing interests

The authors declare no competing interests.
