## [Peer Review File · Nature Communications]

Reviewers' Comments:

Reviewer #1:

Remarks to the Author:

This study investigates the cell tropism of a human norovirus during symptomatic infection of a highly immunocompromised pediatric patient. The findings are largely consistent with a previous study of multiple biopsy patients (Karandikar et al. 2016 J Gen Virol 97(9):2291-2300), showing viral antigen detected in both intestinal epithelial cells and subepithelial cells including macrophages and dendritic cells. One novel observation in this study is the detection of viral negative sense RNA in enteroendocrine cells.

Major concerns:

- 1) The conclusions drawn by the authors are not entirely supported by the data. For example, the authors state repeatedly that negative sense viral RNA was detected nearly exclusively in intestinal epithelial cells. This is not supported by their data (see below). Furthermore, the authors fail to perform quantitative analysis of their data which would provide a more complete picture of the prevalence of specific cell types targeted by a human norovirus.
- 2) The authors draw broad conclusions about norovirus cell tropism from studies of a single severely immunocompromised patient. Acknowledging the inherent difficulties in obtaining human tissue, it is understandable that a large sample set was not available. However, this does not diminish the need to be cautious and narrow about interpreting results from $n = 1$. The authors should emphasize this point in the manuscript because it could significantly alter the types and functional state of the cell types present in this tissue biopsy compared to the normal cellular milieu which could in turn impact the tropism of a virus infecting in this environment.

Specific comments:

- 1) The manuscript presents a biased view of human norovirus cell tropism. While the authors cite four studies that reported detection of viral antigen in immune cells in their introduction, they seem to dismiss these as relevant targets but no explanation for this dismissal is given. Similarly, in their discussion they state that "Although viral antigen was abundant in the lamina propria during acute infection, we could not demonstrate compelling evidence for active replication in immune cells." Yet they detected abundant VP1 protein and positive sense viral RNA, and less abundant but still convincing levels of negative sense viral RNA (Fig. 3 and Extended data Fig. 4) and viral nonstructural protein (Extended data Fig. 3), in subepithelial cells. It is unclear why these observations are not compelling yet detection of the same viral replication intermediates are compelling evidence for epithelial cell infection.

- 2) The authors fail to quantify the number of infected cells in any of their experiments. One assumes they selected representative images for presentation, but quantification across multiple fields of view and across serial sections stained independently will substantially increase confidence in the authors' conclusions.
- 3) Based on the images presented in Fig. 2D-F and Extended data Fig. 3, I am not convinced that viral negative sense RNA and nonstructural proteins were localized predominantly in epithelial cells. Fig. 2 images show few cells positive for negative sense RNA and it is unclear that a majority are epithelial cells. In panel 2D, there is no clear villous so it is difficult to interpret tissue architecture. In panel 2E, there are three positive cells, one of which is underneath the epithelium suggesting it is an immune or stromal cell. Similarly, in Extended data Fig. 3, there appear to be cells positive for nonstructural proteins in the lamina propria but without quantification it is hard to fully glean their prevalence compared to epithelial cell-positive cells.
- 4) I have the same concern with Fig. 3 and Extended data Fig. 4 – there appear to be similar numbers of epithelial and subepithelial cells positive for negative strand in multiple representative images although the authors state that signal is predominantly in epithelial cells.
- 5) It is interesting that the authors found evidence for infection of enteroendocrine cells which has not previously been reported. It would be informative to describe the frequency of this infection event. The authors state that there were virus-negative CgA-positive cells and virus-positive CgA-negative cells. What were the frequencies of each of these? In other words, what proportion of enteroendocrine cells are infected by the virus and what proportion of virus-positive cells are enteroendocrine cells in this patient? Considering the title of the manuscript and the significance presented in the abstract pertain to this finding, these data should be analyzed in a more rigorous manner. The same applies to M cells and tuft cells – how many M/tuft cells were visualized and what percentage of those were positive for negative sense RNA?
- 6) In Extended Data Fig. 6, the authors test for co-localization of VP1, a macrophage marker, and cytokeratin. They conclude this pattern is "consistent with active scavenger activity during the acute infection." However, they also show that this is not a uniform observation since VP1 is detected in the lamina propria in the absence of abundant cytokeratin (Extended Data Fig. 7). Moreover, lamina propria macrophages continuously function to phagocytose apoptotic debris from terminally differentiated epithelial cells even in the absence of viral infection. Detecting cytokeratin in an intestinal macrophage does not prove absence of viral infection when viral antigens are detected inside the cell, nor does it prove that the viral antigens were acquired by phagocytosis instead of infection. Yet the title of this subsection is "Dendritic cells and macrophages process large amounts of norovirus antigen during acute infection." This is misleading and not indicated by the data.
- 7) The B and T cell data included in the manuscript are not sufficient to draw the

conclusion that these cell types do not support active viral replication. First, they did not show results from co-staining of a B cell marker and viral antigen or viral genomes. Second, while the data presented in Fig. 6 D-E suggest that VP1 protein is frequently adjacent to, but not co-localized with, T cells markers, these low-magnification images are insufficient to conclusively rule this out. Moreover, why did the authors not test viral genome localization in B and T cells like they did for other cell types? The authors should also discuss the observation here of inapparent T cell infection versus the findings in Karandikar et al. where viral antigen was frequently detected in T cells. Finally, can the authors discuss the implications of this patient being on lymphocyte inhibitors (tacrolimus and mycophenolate) at the time of illness and endoscopy in relation to their observations?

Reviewer #2:

Remarks to the Author:

The manuscript titled "Human Norovirus Targets Enteroendocrine Epithelial Cells in the Small Intestine" describes the cellular tropism in the gastrointestinal tract of a human norovirus in an acutely infected pediatric patient. Using unique tissue samples obtained from sectional biopsies, Green et al demonstrate norovirus capsid antigen and positive sense RNA in both the epithelium as well as the lamina propria of the small intestine but not in the colon. They extend this finding by determining that active viral replication (detected by the negative sense RNA) was localized predominately to the epithelium or near the epithelium with little to no staining in the lamina propria. This finding was confirmed by detection of non-structural proteins in the same locations. Using cell type specific identifiers, the associate expression of enteroendocrine cell markers and negative sense RNA was found. They see no association of capsid proteins or viral RNA with either CD3 T cells or CD20 B cells. They provide interesting and novel assessments, using confocal microscopy, of the intracellular distribution patterns of structural, non-structural, positive and negative sense RNAs in the infected cells. This is a very carefully performed and well written study in which all reagents were validated in several ways including the use of intestinal biopsies from norovirus-negative patients. The quality of the data and images is exceptional. The detection of the viral genome is new and was an important tool in this study and suggests that detection of viral antigen in cells in the lamina propria does not automatically indicate that viral replication is occurring in the cell. The conclusion of this study have significant implications for defining the pathogenesis of norovirus. The pathogenesis of human norovirus has remained mostly elusive since the identification of the virus mainly due to the lack of relevant clinical specimens in which the questions surrounding pathogenesis could be addressed. There is only one paper at this time that has looked at norovirus in intestinal biopsies from humans and stained for cell types.

Karandikar et al (2016) examined biopsies from immunocompromised patients in which norovirus is a chronic infection and was able to detect norovirus capsid proteins in enterocytes, macrophages, T cells, and dendritic cells while non-structural proteins indicated replication only in enterocytes (Karandikar et al 2016). B cells are also thought to be a target for the virus based on infection of in vitro transformed human cell lines (Jones et al 2014). Recent work using untransformed human intestinal organoid cultures has implied that the differentiated enterocyte is the target of human norovirus (Ettayebi et al 2016). In the murine norovirus model, in which the disease differs substantially from the human situation, M cells, macrophages, dendritic cells, B cells, T cells, and tuft cells have been shown to be infected instead of enterocytes or enteroendocrine cells (Grau et al 2017, Gonzalez-Hernandez et al 2015, Wilen et al 2018). Thus, the true tropism of human norovirus as it commonly occurs in acute infection of the human host is still a matter of much debate. Given this ongoing debate, publication of well conducted studies that look at different proposed cell types that support norovirus replication will provide clear answers that are critical to move studies on norovirus biology forward. This report by Green et al confirms and significantly extends the previously published work of Karandikar et al in chronically infected patients by clearly demonstrating both the enteroendocrine cell and the enterocyte exhibit evidence of infection and viral replication during acute infection associated with norovirus disease. This work provides a significant piece of information that will change the course of the debate on the target of norovirus replication. The association of norovirus with the enteroendocrine cell is novel and the relevance of this finding remains to be determined. Overall, this manuscript provides valuable insight into interactions between norovirus and the cells of the human intestine and is an exciting step in developing targets to either treat or prevent the infection.

Minor Comments:

- It would be great to have included co-staining for negative sense RNA or non structural proteins along with the dendritic cell, macrophage, B and T cell markers to really demonstrate limited/no viral replication was occurring in these cell populations
- This reviewer realizes the difficulty of obtaining the clinical sample during acute infection but additional patient samples would provide more confidence in the results.
- Was there any attempt to determine exactly what type of enteroendocrine cells was infected? Clearly not all are infected including enteroendocrine cells close to each other so is there something special about the infected enteroendocrine cell?

Reviewer #3:

Remarks to the Author:

Green and colleagues performed a careful analysis of intestinal biopsies obtained from a patient who was status/post small intestinal transplantation and who subsequently became infected with a GII.4 Sydney/2012-like virus. They performed stains using capsid-specific, noroviral non-structural protein-specific, and positive- and negative-sense viral RNA specific probes, as well as probes targeting various cell-specific proteins. In some instances confocal microscopy was used to capture images. The authors found evidence of replication (expression of non-structural proteins, presence of negative strand RNA) occurred primarily in enterocytes but not in immune cells.

A major question in the field has been whether murine norovirus is a representative model of human norovirus infection, including which types of cells are infected in vivo. The work by Green and colleagues builds on previous data to clearly document the presence of viral proteins and positive and negative strands of RNA in epithelial cells, including enterocytes and (for the first time) enteroendocrine cells. Capsid protein and positive sense viral RNA, potentially representing virus, are also found in the lamina propria, but negative sense RNA and non-structural proteins were either not observed or less clearly present (extended data figure 3E) in the lamina propria. These data provide strong support to the conclusion that human norovirus is distinct in its in vivo tropisms from that of murine norovirus.

A few points should be addressed to strengthen the manuscript:

Many enterocytes have only the VP1 capsid or positive sense genome present. If these findings don't represent active infection, how do these protein or +RNA get into the cell in sufficient amounts to stain? Is the negative sense RNA detection as sensitive as detection of positive sense RNA and or viral VP1? If less sensitive, what implications does this have for failure to detect evidence of replication in immune cells? Consider adding this as a limitation of the study.

It would be worthwhile to show a video file of CD20-positive cells and VP1-stained tissue, as is done for CD4-positive cells in video 3. Figures 6A and 6B suggest the possibility that some B cells within the GALT also are VP1-antigen positive. This should be clarified, especially if the findings are similar to those observed with the CD4 staining.

Figure 1 – the bar in each panel represents what length? The size the bar represents should be noted in all figures, as is done in Figure 2.

Figure 2 – the DAPI staining looks to be more purple than blue.

Figure 2c – the orientation of the tissue is difficult to ascertain. Is the lumen at the upper left and do the intensely staining (red) cells represent enterocytes (appears to be from the text)? Please note the site of the lumen for clarity.

Figure 3B – is the -RNA at the bottom of the panel on the other side of the villus? It is not clear whether it is associated with the epithelium or the lamina propria. Please clarify.

Line 269 – it is not clear from the figures shown that the negative sense RNA is perinuclear.

Supplementary Figure 2 legend – “magnified insets of B. and C. mark a CgA-positive cell proximal to epithelial cells containing” Do the authors mean ‘proximate to epithelial cells’ (i.e., near such cells)?

I could not get supplementary video files 1 and 3 to play.

Reviewer #1 (Remarks to the Author):

This study investigates the cell tropism of a human norovirus during symptomatic infection of a highly immunocompromised pediatric patient. The findings are largely consistent with a previous study of multiple biopsy patients (Karandikar et al. 2016 J Gen Virol 97(9):2291-2300), showing viral antigen detected in both intestinal epithelial cells and subepithelial cells including macrophages and dendritic cells. One novel observation in this study is the detection of viral negative sense RNA in enteroendocrine cells.

Major concerns:

1) The conclusions drawn by the authors are not entirely supported by the data. For example, the authors state repeatedly that negative sense viral RNA was detected nearly exclusively in intestinal epithelial cells. This is not supported by their data (see below). Furthermore, the authors fail to perform quantitative analysis of their data which would provide a more complete picture of the prevalence of specific cell types targeted by a human norovirus.

Response: This is a serious comment, and it prompted us to exhaustively re-examine our existing images, as well as to perform additional experiments with new tissue recuts from patient GT-1 and others. Our finding that human norovirus targets enteroendocrine cells (EECs) is reproducible in other patients, and we will include a quantitative estimate of EEC target cell infection in intestinal tissue. The reviewer's challenge to our conclusions about the susceptibility of immune cells to norovirus infection led us to re-examine patient samples with additional high-resolution imaging. Our experimental approach is outlined below in the specific comments section.

2) The authors draw broad conclusions about norovirus cell tropism from studies of a single severely immunocompromised patient. Acknowledging the inherent difficulties in obtaining human tissue, it is understandable that a large sample set was not available. However, this does not diminish the need to be cautious and narrow about interpreting results from $n = 1$. The authors should emphasize this point in the manuscript because it could significantly alter the types and functional state of the cell types present in this tissue biopsy compared to the normal cellular milieu which could in turn impact the tropism of a virus infecting in this environment.

Response: We recognize that $n=1$ is a narrow sample size. To address the reviewer's comment, we have added new supporting data from three additional patients that show an association of human norovirus infection with chromogranin A-positive EECs (new Extended Data Figures 6 and 7). At this time, we have access only to biopsies from immunocompromised patients, whose biopsies were collected as part of their clinical care for an underlying condition. We agree that biopsies from non-immunocompromised patients with acute diarrhea would be instructive, but endoscopy during infectious diarrhea is not routine and recent human challenge studies have not included biopsy sampling. Because this remains an important question in the field, we have recently developed a norovirus natural history protocol that will give us the ability to study biopsies from immunocompetent patients with diarrhea. Although these procedures may be rare (we expect logistics to be somewhat complicated due to the unpredictable timing of illnesses), we will keep pursuing these samples. We had recognized the importance of data from immunocompetent

patients in our submitted paper with the statement:

“Additional study will be needed to establish the role of EECs in **both immunocompetent and immunocompromised** human hosts with norovirus disease, but the wide-ranging roles of EECs in gut-brain signaling, digestion, and mucosal immunity could offer important insight into new strategies to treat norovirus illness.” We will leave this statement in the revised version to address this important point.

Specific comments:

1) The manuscript presents a biased view of human norovirus cell tropism. While the authors cite four studies that reported detection of viral antigen in immune cells in their introduction, they seem to dismiss these as relevant targets but no explanation for this dismissal is given. Similarly, in their discussion they state that “Although viral antigen was abundant in the lamina propria during acute infection, we could not demonstrate compelling evidence for active replication in immune cells.” Yet they detected abundant VP1 protein and positive sense viral RNA, and less abundant but still convincing levels of negative sense viral RNA (Fig. 3 and Extended data Fig. 4) and viral nonstructural protein (Extended data Fig. 3), in subepithelial cells. It is unclear why these observations are not compelling yet detection of the same viral replication intermediates are compelling evidence for epithelial cell infection.

Response: Intestinal tissue can contain abundant VP1 capsid protein, and in fixed tissue it is challenging to distinguish among: 1) incoming virus, 2) VP1 translated from subgenomic RNA during active replication, and 3) progeny virus. It is also difficult to determine whether capsid protein in immune cells is due to active replication in these cells or processing of antigen. Karandikar et al. addressed this problem in human biopsies by scoring replication-positive intestinal cells as those co-expressing both VP1 and nonstructural proteins. In our study, we developed norovirus strain-specific RNAscope probes (ACD) to assess permissive cells. We scored replication-positive cells as those expressing norovirus-specific negative sense RNA. That said, myeloid immune cells might score positive for negative sense RNA or any other product of norovirus replication due to its antigen processing function. Thus, our study emphasized predominant patterns of viral protein and RNA expression (and their distribution) in the tissue. Additional recut tissue slides were acquired from the Georgetown pathology lab and probed further with immune cell markers and confocal imaging to re-evaluate our data. We worked with ACD to improve the detection and resolution of the hybridized RNAscope probes with an Opal conjugate in tandem with immunofluorescence for detection of VP1 or NS7. The company asserts that RNAscope probe signals appear dot-like and intense for each molecule of hybridized RNA, which in our biopsies appeared as intense individual dots or as bright regions of overlapping signals, sometimes with an aggregated appearance. In some views, the negative sense RNA signal appeared perinuclear, and in other views it did not. Red blood cells can appear as dull, reddish “blobs” in the fluorescent RNA probe channel, which is why this paper included a number of classical immunohistochemistry images collected by brightfield microscopy to verify certain patterns of expression. One intense fluorescent negative strand signal image pointed out by the reviewer in Figure 2E (that was indicated with a white arrow) was sliced and reconstructed as a 3D image to show that the negative sense RNA was in close association with epithelial marker CK and not macrophage marker IBA-1. This reconstruction is now included as new Supplementary Video 1. We performed a new series of experiments with negative strand ISH and immune cell markers (IBA-1, CD3, and CD20) so that we could generate additional 3D reconstructions. With this approach we again found a comparatively less abundant signal of norovirus negative strand RNA in the LP as illustrated in the new Panels E and F in Figure 5. The reviewer’s point that we could not distinguish between antigen processing (internalization) and active replication (see comment 6 below) is well-taken. The presence of negative strand RNA in macrophage could be due to phagocytosis, passive internalization of viral replication complexes, or indeed, active replication. And as pointed out by reviewer 3 below, the lower amount of signal from the negative strand RNA probe might simply be due to the lower abundance of this replicative intermediate in the virus replication cycle. We re-examined replication with the T and B cell markers, and include new IF images and video (new Panel G in Figure 6 and Supplementary Video 6) showing the general absence of internal negative sense RNA signals in B and T cells. These data are consistent with the general absence of viral RNA and protein in the inner regions of B and T cell-heavy GALT (Figure 6), which is in contrast to the murine norovirus model. The comments by the reviewer illustrate that our images will be scrutinized closely for inconsistencies in the signals, and we now recognize that certain views can be open to a different interpretation. Taken together, we have decided to address these inconsistencies by noting them and using more cautious language in our description of the immune cell data, acknowledging in the Discussion that we could not determine in this study whether replication might occur in immune cells under certain conditions. As technologies continue to improve, and with the analysis of many more clinical samples, we hope that our work will stimulate further investigation into tropisms for immune cells.

2) The authors fail to quantify the number of infected cells in any of their experiments. One assumes they selected representative images for presentation, but quantification across multiple fields of view and across serial sections stained independently will substantially increase confidence in the authors' conclusions.

Response: We selected images for the original submission that we had determined as representative of the overall pattern of staining for individual probes and markers. Although not shown, we performed several independent experiments for each RNA probe and antibody, as we optimized dilutions and conditions. Images were collected and processed by experienced investigators and data were reviewed by the entire team. When we saw signals that diverged from the predominant pattern (such as those pointed out by Reviewer 1), we processed many additional fields and tissues. When needed, we collected data from confocal microscopy and made cross-sections and reconstructed 3D images to assess co-localization within the same cell. Taken together, our conclusions were determined from predominant patterns within multiple views of different tissues. We have now included data from additional patients to support our conclusion that EECs are target cells. New Figures are included in the revised version that show larger fields of view at lower magnification for better perspective (New Revised Figure 4, Panel A; Figure 5, new panel E; Figure 6, new panel G; New Extended Data 7, panel B). See response to Specific comment 5 below regarding quantification of EECs.

3) Based on the images presented in Fig. 2D-F and Extended data Fig. 3, I am not convinced that viral negative sense RNA and nonstructural proteins were localized predominantly in epithelial cells. Fig. 2 images show few cells positive for negative sense RNA and it is unclear that a majority are epithelial cells. In panel 2D, there is no clear villous so it is difficult to interpret tissue architecture. In panel 2E, there are three positive cells, one of which is underneath the epithelium suggesting it is an immune or stromal cell. Similarly, in Extended data Fig. 3, there appear to be cells positive for nonstructural proteins in the lamina propria but without quantification it is hard to fully glean their prevalence compared to epithelial cell-positive cells.

Response: These comments highlight one of the challenges in working with these small and limited amounts of biopsy samples. The best images result from cutting precisely through the villus in a properly embedded biopsy, but the complex three-dimensional nature of these sections can present technical challenges. We re-visited these concerns as addressed in our response to comment 1 above by requesting more recuts of the archived tissue and performing additional experiments to probe subepithelial cells with the use of high magnification and confocal techniques.

4) I have the same concern with Fig. 3 and Extended data Fig. 4 – there appear to be similar numbers of epithelial and subepithelial cells positive for negative strand in multiple representative images although the authors state that signal is predominantly in epithelial cells.

Response: In our revised version we have added images of viral negative strand RNA and the nonstructural protein NS7^{Pol} together within the cell as stronger evidence of active replication (New Extended Data 4). Such cells displaying both signals were readily visualized in the epithelium. However, negative sense RNA signals within the LP are now acknowledged (see legend to Extended Data Figure 5C).

5) It is interesting that the authors found evidence for infection of enteroendocrine cells which has not previously been reported. It would be informative to describe the frequency of this infection event. The authors state that there were virus-negative CgA-positive cells and virus-positive CgA-negative cells. What were the frequencies of each of these? In other words, what proportion of enteroendocrine cells are infected by the virus and what proportion of virus-positive cells are enteroendocrine cells in this patient? Considering the title of the manuscript and the significance presented in the abstract pertain to this finding, these data should be analyzed in a more rigorous manner. The same applies to M cells and tuft cells – how many M/tuft cells were visualized and what percentage of those were positive for negative sense RNA?

Response: We expected to find an association of human norovirus infection with tuft cells, so the linkage with EECs was unexpected. There are a number of morphological appearances of EECs (“open” and “closed” types) that have been described in the literature and certain EECs may appear “subepithelial” because they are located close to the basal membrane <https://www.ncbi.nlm.nih.gov/pmc/articles/PMC4842178/>. In the revised version, we provide a new lower magnification image that shows CgA-positive EECs (with and without norovirus negative sense RNA) in the epithelial layer of patient GT-1 (New Revised Figure 4, Panel A). We have included evidence for norovirus infection in three additional immunosuppressed patients, two with chronic infection, and one with

acute diarrhea. We have added a table that lists the ratios of CgA-positive cells with and without evidence of norovirus infection that were counted in three fields of view from three patients, with one of these being GT-1 (New Extended Data Fig 7). Consistent with the published literature that EECs represent approximately 1% of the epithelial cell population, we observed that the EECs were a rare epithelial cell subtype in our biopsies. The CgA-positive EECs were present throughout the small intestine and colon, but only a subset of these EECs were positive for the virus. We consulted with an NIH EEC researcher to apply current technologies to characterize this permissive EEC subset, but learned that human EEC subtypes are less well-defined than those of mice. We plan to investigate this question, but it will require the further development of research tools.

We searched extensively for virus-positive tuft cells, but we could not find norovirus-infected tuft cells in multiple fields of several tissue sections from multiple patients. Our supplemental Figure 1 had shown that tuft cells are present in the epithelial layer, but not abundant (similar to the distribution of EECs). We searched colonic biopsies from patients with long-term chronic norovirus infection, but again, failed to detect evidence of virus infection in tuft cells. We will keep searching for norovirus-positive tuft cells in our studies, as it is an important point. We do not yet know if there are strain differences among human noroviruses in this regard as there appear to be for the murine noroviruses.

In regard to M cells, we had exhausted the GT-1 tissue sections that showed the distinct GALT domes (as shown in Figure 1) prior to optimization of our negative sense RNA probe. We did not include M cell data in the submitted paper or the revised version because we lacked clear GALT tissue to visualize infection in the preferred context in multiple patients. We are working on this now, but it is too early to publish our findings, and we want to remain focused on EECs in this report.

6) In Extended Data Fig. 6, the authors test for co-localization of VP1, a macrophage marker, and cytokeratin. They conclude this pattern is “consistent with active scavenger activity during the acute infection.” However, they also show that this is not a uniform observation since VP1 is detected in the lamina propria in the absence of abundant cytokeratin (Extended Data Fig. 7). Moreover, lamina propria macrophages continuously function to phagocytose apoptotic debris from terminally differentiated epithelial cells even in the absence of viral infection. Detecting cytokeratin in an intestinal macrophage does not prove absence of viral infection when viral antigens are detected inside the cell, nor does it prove that the viral antigens were acquired by phagocytosis instead of infection. Yet the title of this subsection is “Dendritic cells and macrophages process large amounts of norovirus antigen during acute infection.” This is misleading and not indicated by the data.

Response: The reviewer is correct that antigen processing was not experimentally verified in the macrophage and DCs of this study. We changed the title of this section to “Dendritic cells and macrophage **contain** abundant norovirus antigen during acute infection.” In our studies, we consistently found cytokeratin within VP1-positive macrophage in the majority of our 3D reconstructions, suggesting that phagocytosis of infected epithelial cell components was an important host response. We had included Figure Extended Data 7 (now Extended Data Figure 10) in the original submission because this image suggested that phagocytosis of cellular debris might not be the only means of spread in the LP. To confirm active replication in cells of the LP, we used the detection of negative strand RNA as our “gold standard.” We have added additional fluorescent images of sections hybridized with the negative RNA probe and IBA-1 (New Panels E and F in Figure 5). We have modified the closing text of the paper to read as follows:

“Human norovirus is internalized into myeloid cells in the lamina propria by phagocytosis of debris from infected epithelial cells or by lysis-independent spread, perhaps via exosomes as proposed recently³⁶. Whether internalized human norovirus remains viable and infectious within myeloid immune cells for a period of time or replicates under certain conditions **could not be determined in our study**, but it is likely that gut homeostasis is affected by the presence of large numbers of activated antigen-presenting cells, especially during an acute infection.”

7) The B and T cell data included in the manuscript are not sufficient to draw the conclusion that these cell types do not support active viral replication. First, they did not show results from co-staining of a B cell marker and viral antigen or viral genomes. Second, while the data presented in Fig. 6 D-E suggest that VP1 protein is frequently adjacent to, but not co-localized with, T cells markers, these low-magnification images are insufficient to conclusively rule this out. Moreover, why did the authors not test viral genome localization in B and T cells like they did for other cell types? The authors should also discuss the observation here of inapparent T cell infection versus the findings in Karandikar et al. where viral antigen was frequently detected in T cells. Finally, can the

authors discuss the implications of this patient being on lymphocyte inhibitors (tacrolimus and mycophenolate) at the time of illness and endoscopy in relation to their observations?

Response: In order to address the possibility that we overlooked replication in B and T cells, we re-examined the biopsies (as outlined in the response to comment 1 above) with confocal microscopy. We have now included new high-resolution images of cells probed with B and T cell markers and hybridized with the negative strand RNA probe (Revised Figure 6 with new Panel G). The figure includes panels with the immune cell markers alone, the negative strand RNA probe alone, and the merged images. B and T cells were readily detected throughout the LP, showing their presence in this immunosuppressed patient. We could not detect co-localization of the T and B cell markers with negative sense RNA, and as consistently observed throughout our studies, negative strand RNA was not abundant in the lamina propria. We cannot explain the discrepancy in T cell data with the Karandikar et al. paper, but they did not include 3D imaging that would have revealed whether the abundant viral antigen they detected was definitely within the T cells and not adjacent (as we showed in Video 3 of our original submission, that is now Video 5). We have added another video showing that signals from proximal T and B cells did not co-localize with negative strand RNA (new Video 6 in revised version).

In regard to the immunosuppression of patient GT-1, we now include clinical data that the patient's T cell numbers were within normal range at the time of illness. Thus, there were normal numbers of T cells available to serve as target cells, assuming the T cells were in a permissive state. According to Dr. Stuart Kaufman, the patient was in a "trough" of his immunosuppressive therapy, which had been reduced to minimal maintenance levels (after an initial high dosage) to prevent rejection of his transplanted organs.

Reviewer #2 (Remarks to the Author):

The manuscript titled "Human Norovirus Targets Enteroendocrine Epithelial Cells in the Small Intestine" describes the cellular tropism in the gastrointestinal tract of a human norovirus in an acutely infected pediatric patient. Using unique tissue samples obtained from sectional biopsies, Green et al demonstrate norovirus capsid antigen and positive sense RNA in both the epithelium as well as the lamina propria of the small intestine but not in the colon. They extend this finding by determining that active viral replication (detected by the negative sense RNA) was localized predominately to the epithelium or near the epithelium with little to no staining in the lamina propria. This finding was confirmed by detection of non-structural proteins in the same locations. Using cell type specific identifiers, the associate expression of enteroendocrine cell markers and negative sense RNA was found. They see no association of capsid proteins or viral RNA with either CD3 T cells or CD20 B cells. They provide interesting and novel assessments, using confocal microscopy, of the intracellular distribution patterns of structural, non-structural, positive and negative sense RNAs in the infected cells. This is a very carefully performed and well written study in which all reagents were validated in several ways including the use of intestinal biopsies from norovirus-negative patients. The quality of the data and images is exceptional. The detection of the viral genome is new and was an important tool in this study and suggests that detection of viral antigen in cells in the lamina propria does not automatically indicate that viral replication is occurring in the cell. The conclusion of this study have significant implications for defining the pathogenesis of norovirus. The pathogenesis of human norovirus has remained mostly elusive since the identification of the virus mainly due to the lack of relevant clinical specimens in which the questions surrounding pathogenesis could be addressed. There is only one paper at this time that has looked at norovirus in intestinal biopsies from humans and stained for cell types. Karandikar et al (2016) examined biopsies from immunocompromised patients in which norovirus is a chronic infection and was able to detect norovirus capsid proteins in enterocytes, macrophages, T cells, and dendritic cells while non-structural proteins indicated replication only in enterocytes (Karankikar et al 2016). B cells are also thought to be a target for the virus based on infection of in vitro transformed human cell lines (Jones et al 2014). Recent work using untransformed human intestinal organoid cultures has implied that the differentiated enterocyte is the target of human norovirus (Ettayebi et al 2016). In the murine norovirus model, in which the disease differs substantially from the human situation, M cells, macrophages, dendritic cells, B cells, T cells, and tuft cells have been shown to be infected instead of enterocytes or enteroendocrine cells (Grau et al 2017, Gonzalez-Hernandez et al 2015, Wilen et al 2018). Thus, the true tropism of human norovirus as it commonly occurs in acute infection of the human host is still a matter of much debate. Given this ongoing debate, publication of well conducted studies that look at different proposed cell types that support norovirus replication will provide clear answers that are critical to move studies on norovirus biology forward. This report by Green et al confirms and significantly extends the previously published work of Karandikar et al in chronically infected patients by clearly demonstrating both the enteroendocrine cell and the enterocyte exhibit evidence of infection and viral replication during acute infection associated with norovirus disease. This work provides a significant piece of information that will change the course of the debate on the target

of norovirus replication. The association of norovirus with the enteroendocrine cell is novel and the relevance of this finding remains to be determined. Overall, this manuscript provides valuable insight into interactions between norovirus and the cells of the human intestine and is an exciting step in developing targets to either treat or prevent the infection.

Response: We appreciate the reviewer's comments.

Minor Comments:

- It would be great to have included co-staining for negative sense RNA or non structural proteins along with the dendritic cell, macrophage, B and T cell markers to really demonstrate limited/no viral replication was occurring in these cell populations

Response: Thank you for this suggestion. We have now included negative strand RNA ISH with the B and T cell markers in a new Panel G in Figure 6. We also include new images with co-detection of the viral polymerase protein and negative sense RNA within the same epithelial cell (New Extended Data 4).

- This reviewer realizes the difficulty of obtaining the clinical sample during acute infection but additional patient samples would provide more confidence in the results.

Response: We have added data from three additional patients enrolled in the two IRB-approved studies included in this report.

- Was there any attempt to determine exactly what type of enteroendocrine cells was infected? Clearly not all are infected including enteroendocrine cells close to each other so is there something special about the infected enteroendocrine cell?

Response: This is an exciting area of future research and we have established interaction with an EEC research team here at NIH. The human EEC field lags behind that of the murine EEC field, which has reporter mice for marking different EEC subtypes that have been well-defined in the animal. Chromogranin A detection is an accepted broadly reactive marker for EECs, but the identity of specific human EEC subclasses involved in norovirus replication will need additional study and we will pursue this with our NIH collaborators.

Reviewer #3 (Remarks to the Author):

Green and colleagues performed a careful analysis of intestinal biopsies obtained from a patient who was status/post small intestinal transplantation and who subsequently became infected with a GII.4 Sydney/2012-like virus. They performed stains using capsid-specific, noroviral non-structural protein-specific, and positive- and negative-sense viral RNA specific probes, as well as probes targeting various cell-specific proteins. In some instances confocal microscopy was used to capture images. The authors found evidence of replication (expression of non-structural proteins, presence of negative strand RNA) occurred primarily in enterocytes but not in immune cells.

A major question in the field has been whether murine norovirus is a representative model of human norovirus infection, including which types of cells are infected in vivo. The work by Green and colleagues builds on previous data to clearly document the presence of viral proteins and positive and negative strands of RNA in epithelial cells, including enterocytes and (for the first time) enteroendocrine cells. Capsid protein and positive sense viral RNA, potentially representing virus, are also found in the lamina propria, but negative sense RNA and non-structural proteins were either not observed or less clearly present (extended data figure 3E) in the lamina propria. These data provide strong support to the conclusion that human norovirus is distinct in its in vivo tropisms from that of murine norovirus.

Response: We appreciate the comments of the reviewer.

A few points should be addressed to strengthen the manuscript:

Many enterocytes have only the VP1 capsid or positive sense genome present. If these findings don't represent active infection, how do these protein or +RNA get into the cell in sufficient amounts to stain? Is the negative sense

RNA detection as sensitive as detection of positive sense RNA and or viral VP1? If less sensitive, what implications does this have for failure to detect evidence of replication in immune cells? Consider adding this as a limitation of the study.

Response: The uneven VP1 and positive strand RNA signals in enterocytes noted by the reviewer were the subject of many discussions by our group. One explanation could be technical. In our experience with these tissues and 3D reconstructions, we have learned that there can sometimes be a predominant signal in a cell that obscures a second signal during image collection and processing. Moreover, the RNAscope probes have an amplification mechanism that was extremely intense in some images. In addition, the cutting process can damage tissue morphology or remove a section of a cell. Another explanation is that the observed pattern reflects internalization or mechanistic spread of the virus or its components in the enterocytes (or other cells) via processes that we don't yet understand. Finally, abundant signals in the enterocytes could indeed reflect active replication, consistent with our proposal from many data points that replication occurs in epithelial cells. Our revised version includes new data and additional figures that give better views of the overall picture of an acute norovirus infection in action, and admittedly, we still have more to learn. Based on the known principles of calicivirus replication, we would expect the amount of negative strand RNA to be much lower than that of the positive strand RNA. Indeed, this difference in abundance could lead to differences in the efficiency of detection by our current tools. We have added a reference to address this caveat with the following statement: "The viral negative sense RNA signal was less abundant overall, likely reflecting its lower copy number during the viral replication cycle <https://www.ncbi.nlm.nih.gov/pubmed/22626565> (now reference 22). The strongest negative strand signals were found in or near the epithelial layer (Fig. 2D and E). The distribution pattern and lower intensity of nonstructural protein expression was similar to that of the negative strand RNA, with the strongest signals from nonstructural proteins NS5^{VPg}, NS6^{Pro}, and NS7^{Pol} detected in or near the epithelium (Extended Data Fig. 3A-C and E). Confocal imaging confirmed the presence of both NS7^{Pol} and negative sense RNA within epithelial cells, consistent with active replication and we have added a new figure that shows this (new Extended Data Fig. 4)."

It would be worthwhile to show a video file of CD20-positive cells and VP1-stained tissue, as is done for CD4-positive cells in video 3. Figures 6A and 6B suggest the possibility that some B cells within the GALT also are VP1-antigen positive. This should be clarified, especially if the findings are similar to those observed with the CD4 staining.

Response: We agree that Figure 6 made it difficult to see potential co-localization with the T and B cell markers at the periphery of the GALT. We have added new magnified fluorescent images with B and T cell marker staining alone or in the presence of negative strand probe and then merged (new Panel G of Figure 6 and New Supplementary Video 6). In spite of an intensive search for evidence of infection, the dissociation of the B cell marker and negative strand RNA was clear-cut, and we were unable to generate a video where it might be questionable.

Figure 1 – the bar in each panel represents what length? The size the bar represents should be noted in all figures, as is done in Figure 2.

Response: The size marker definition, which was the same for all the images, was added to the figure legend, as well as a number of the other figures.

Figure 2 – the DAPI staining looks to be more purple than blue.

Response: The DAPI color was assigned by the computer software that we used. It appears blue on our monitors, and we apologize that it appears purple on others. If our paper is accepted, we will work with the journal to ensure that the colors are accurate in the published version.

Figure 2c – the orientation of the tissue is difficult to ascertain. Is the lumen at the upper left and do the intensely staining (red) cells represent enterocytes (appears to be from the text)? Please note the site of the lumen for clarity.

Response: We have added this label to the panel (Figure 2C).

Figure 3B – is the -RNA at the bottom of the panel on the other side of the villus? It is not clear whether it is associated with the epithelium or the lamina propria. Please clarify.

Response: To clarify, we brightened the image and we believe that these were likely overlapping epithelial layers due to the alignment of the nuclei. However, we did not have cellular markers in this figure, unfortunately. Hopefully, there will be enough redundancy in other data to address this problem.

Line 269 – it is not clear from the figures shown that the negative sense RNA is perinuclear.

Response: Indeed, this magnification does not show the intracellular distribution clearly. We removed this description.

Supplementary Figure 2 legend – “magnified insets of B. and C. mark a CgA-positive cell proximal to epithelial cells containing” Do the authors mean ‘proximate to epithelial cells’ (i.e., near such cells)?

Response: This has been clarified in the figure and figure legend.

I could not get supplementary video files 1 and 3 to play.

Response: We are sorry about this, as we submitted the files in an approved format for the journal. We are happy to provide new video files for review. Our revised version includes additional videos and we trust that they will be functional.

We would like to thank the reviewers for their excellent suggestions. The manuscript has been fortified with new data and additional patient samples. Our hope is that this work will be acceptable for publication in Nature Communications so that it can be shared via open access and stimulate further investigation. Thank you for your consideration.

Sincerely,

Kim Y. Green, Ph.D.
Sr. Investigator
Chief, Caliciviruses Section
Laboratory of Infectious Diseases
NIAID/National Institutes of Health
Department of Health and Human Services
Building 50, Room 6318
50 South Drive-MSB8026
Bethesda, Maryland 20892
Phone: 301 908 2721
kgreen@niaid.nih.gov

Reviewers' Comments:

Reviewer #1:

Remarks to the Author:

I appreciate the authors' thoughtful and detailed response to my previous concerns about this study. I still have reservations with their assertion that "human noroviruses preferentially target epithelial cells" based on these data collected from a very small sample size of immunocompromised people. The data for EEC infection are novel and convincing, and should open up new lines of investigation into how noroviruses cause disease.

Reviewer #2:

Remarks to the Author:

The authors have more than adequately addressed the concerns of the reviewers and have greatly extended their already significant findings with the inclusion of additional data. This reviewer has no further comments.

Reviewer #3:

Remarks to the Author:

The authors have answered my questions satisfactorily. There are a couple of additional clarifications they can make as noted below:

Supplementary video 1 legend – please provide a legend for the various colors in the video.

Extended data Fig 4 – please provide orientation to help the reader understand how cells identified as epithelial cells

Reviewer #1 (Remarks to the Author):

I appreciate the authors' thoughtful and detailed response to my previous concerns about this study. I still have reservations with their assertion that "human noroviruses preferentially target epithelial cells" based on these data collected from a very small sample size of immunocompromised people. The data for EEC infection are novel and convincing, and should open up new lines of investigation into how noroviruses cause disease.

Author Response: We thank the reviewer for these comments, and the critique that strengthened our manuscript. We have added further cautionary language in regard to the small sample size and immunocompromised state of these patients as follows:

"Previous observations, coupled with our strand-specific viral RNA analysis presented here (**albeit in a limited number of immunocompromised patients**), indicate that intestinal epithelial cells in the small intestine are major primary target cells for human norovirus replication (Supplementary Fig. 3)."

Reviewer #2 (Remarks to the Author):

The authors have more than adequately addressed the concerns of the reviewers and have greatly extended their already significant findings with the inclusion of additional data. This reviewer has no further comments.

Author Response: We thank the reviewer for constructive comments through this review process.

Reviewer #3 (Remarks to the Author):

The authors have answered my questions satisfactorily. There are a couple of additional clarification they can make as noted below:

Supplementary video 1 legend – please provide a legend for the various colors in the video.

Author Response: Thank you for your review and for noting this oversight. We have provided the color description to the Supplementary video 1 legend.

Extended data Fig 4 – please provide orientation to help the reader understand how cells identified as epithelial cells

Author Response: In the absence of cytokeratin staining in this experiment, we have modified the figure to outline cells morphologically consistent with the epithelial layer.

We appreciate this opportunity to further improve our manuscript. Please do not hesitate to request clarification on our response to this second review.